Pelvic floor muscle training associated with the photobiomodulation therapy for women affected by the genitourinary syndrome of menopause: a study protocol

Oliveira Bezerra Lívia 1 4
de Carvalho Maria Letícia Araújo Silva 2
Silva-Filho Edson 3 emailmeneses@gmail.com
Clara Eugênia de Oliveira Maria 2
http://orcid.org/0000-0003-4595-6746 de Andrade Palloma Rodrigues 4
Micussi Maria Thereza Albuquerque Barbosa Cabral 4
1 Department of Physiotherapy, Federal University of Rio Grande do Norte , Natal, Rio Grande do Norte , Brazil
2 Department of Phisiotherapy, Federal University of Rio Grande do Norte , Natal, Rio Grande do Norte , Brazil
3 Department of Physical Therapy, Universidade Federal do Rio Grande do Norte , Natal, Rio Grande do Norte , Brazil
4 Department of Physical Therapy, Federal University of Rio Grande do Norte , Natal, Rio Grande do Norte , Brazil
Oliveira Sonia
Electronic publication date: 2024 Nov 29
Publication date: 2024
Volume: 12
Electronic Location ID: e17848
Received 2024 Jan 19; Accepted 2024 Jul 10
Copyright: © 2024 Oliveira Bezerra et al.
Copyright year: 2024
Copyright holder: Oliveira Bezerra et al.
License: This is an open access article distributed under the terms of the Creative Commons Attribution License, which permits unrestricted use, distribution, reproduction and adaptation in any medium and for any purpose provided that it is properly attributed. For attribution, the original author(s), title, publication source (PeerJ) and either DOI or URL of the article must be cited.
License URL: https://creativecommons.org/licenses/by/4.0/

Keywords: Pelvic floor disorders, Low-level light therapy, Sexual dysfunction

Funding: Coordenação de Aperfeiçoamento de Pessoal de Nível Superior-Brasil (CAPES)–Code 001 This study was funded by the Coordenação de Aperfeiçoamento de Pessoal de Nível Superior-Brasil (CAPES)–Code: 001. The funders had no role in study design, data collection and analysis, decision to publish, or preparation of the manuscript.

==============================
Background

Frequently, the women affected by the genitourinary syndrome of menopause experience genitourinary dysfunctions that profoundly influence their overall health. Even though the symptoms do not jeopardize the women’s lives, the urinary and sexual dysfunctions significantly impact their quality of life. Isolated treatments focused on the main causes of the dysfunctions, such as pelvic floor muscle training (PFMT) and photobiomodulation have shown significant improvements in genitourinary dysfunctions. So, the association of PFMT with photobiomodulation may generate additional effects in the genitourinary area. This study aims to create a PFMT protocol isolated and associated with photobiomodulation therapy in women affected by the genitourinary syndrome of menopause.

Methods

It is a randomized, controlled, double-blind clinical trial protocol study that will include women experiencing genitourinary symptoms related to menopause, sexually active, never practiced pelvic floor muscle exercise or photobiomodulation treatment, and do not use hormone replacement therapy for at least 3 months. The randomization will allocate the women to three groups: PFMT group, PFMT associated with active intracavitary photobiomodulation group, and PFMT associated with sham intracavitary photobiomodulation group. A total of 16 sessions will be conducted twice a week. The assessments will occur before interventions, after the sixteenth session, and 1 month after the sixteenth session (follow-up). The evaluation will include the pelvic floor muscle vaginal manometry as the primary outcome. Also, the international consultation on incontinence questionnaire—short form, the female sexual function index, the Utian Quality of Life scale, the patient global impression of improvement, the modified Oxford scale, and the vaginal health index will be the secondary outcomes.

Discussion

Despite there are gold standard treatments such as PFMT, to alleviate genitourinary symptoms, interventions mirroring clinical practice are needed. This study protocol might show a groundbreaking and viable method to potentiate the effects of a gold-standard treatment associated with photobiomodulation.

Conclusion

We expect this protocol to demonstrate that the use of PFMT and photobiomodulation strategies is feasible and able to potentiate the recovery of women affected by the genitourinary syndrome of menopause. The Ethics Committee of the Federal University of Rio Grande do Norte approved the study (n° 6.038.283), and the clinical trials platform registered the protocol (n° RBR-5r7zrs2).

Introduction

The genitourinary syndrome of menopause (GSM) is characterized by the presence of signs and symptoms generated from estrogen deficiency in the female genitourinary tract, involving the vagina, labia, vestibule, urethra, and bladder (Portman, Gass & Vulvovaginal Atrophy Terminology Consensus Conference Panel, 2014). Approximately 40% to 50% of women in physiological menopause experience symptoms associated with the GSM (Portman, Gass & Vulvovaginal Atrophy Terminology Consensus Conference Panel, 2014; Palacios et al., 2017). The main signs and symptoms include vaginal atrophy and dryness, vulvovaginal discomfort, and sexual difficulties, such as lack of lubrication and dyspareunia (Portman, Gass & Vulvovaginal Atrophy Terminology Consensus Conference Panel, 2014). Furthermore, the GSM is associated with atrophic changes in the vaginal mucosa, reduction of Lactobacillus levels in the vaginal microbiota, and urinary dysfunctions, significantly impacting women’s overall health and quality of life (Chedraui et al., 2012).

Pelvic floor muscle training (PFMT) is a recommended strategy to treat the GSM (Abrams et al., 2010). There is evidence demonstrating that pelvic floor muscle training improves corporal conscience and strengthens the pelvic muscles, decreasing urinary dysfunctions and enhancing desire, excitement, and orgasm, which are important in female sexual activity (Rosenbaum, 2005). Recently, photobiomodulation has emerged as a therapeutic approach to treat common dysfunctions in the genitourinary syndrome of menopause (Frederice et al., 2022; De Marchi et al., 2023). Some experimental studies showed that the photobiomodulation therapeutic effect is related to photo absorption from specific tissues, such as chromophores (Karu & Kolyakov, 2005). Also, photobiomodulation can induce the expression of regulatory proteins from the cell cycle, activating satellite cells, promoting angiogenesis, increasing the quantity and density of the regenerator fibers, and stimulating mitochondrial activity (Ferraresi et al., 2011; Iyomasa et al., 2009).

Considering these assumptions, photobiomodulation has been utilized to prevent muscle damage, repair muscle tissues, improve muscle performance, and reduce fatigue (Xu et al., 2018; Toma et al., 2016; Ferraresi et al., 2015; Lanferdini, Krüger & Baroni, 2018). Moreover, the chronic use of photobiomodulation associated with strength exercise training potentiates the strengths generated by the exercise (Ferraresi, Huang & Hamblin, 2016; Vanin et al., 2016; Baroni et al., 2014; Ferraresi et al., 2011). Evidence demonstrates that the photobiomodulation intervention before or after an 8-week muscular training potentiated the muscular thickness and strength (Ferraresi, Huang & Hamblin, 2016; Vanin et al., 2016; Baroni et al., 2014; Ferraresi et al., 2011; Vassão et al., 2018). The therapeutic effects generated by photobiomodulation have been associated with improvements in fatigue reduction, post-exercise recovery, and muscular performance (Baroni et al., 2014; Ferraresi et al., 2011; Miranda et al., 2016).

Although pelvic floor alterations, such as urinary and sexual dysfunctions do not threaten women’s lives, they generate significant limitations (Levine, Williams & Hartmann, 2008). In this sense, urinary and sexual dysfunctions impose physical, social, and occupational restrictions, affecting the women’s quality of life (Levine, Williams & Hartmann, 2008). Therefore, it is important to investigate the effects of innovative therapies to treat the dysfunctions generated by the GSM in order to improve women’s health and quality of life.

Considering the photobiomodulation physiological effects on the pelvic floor muscle and the scarcity of clinical trials related to this subject, this preliminary study aims to explore the feasibility and therapeutic window for PFMT isolated and associated with photobiomodulation in women affected by GSM who present urinary and sexual dysfunctions.

Methods

Study setting

It is a randomized, controlled, double-blind clinical trial protocol study that will be performed at the urogynecology outpatient clinic located in the Maternity Hospital Januário Cicco School from the Federal University of Rio Grande do Norte, Natal, Brazil, in 2024.

Eligibility criteria

We will recruit women to participate in this study according to the following criteria: (1) age between 45 and 65 years old; (2) natural menopause at least 1 year ago; (3) no use of hormone replacement therapy for at least 3 months; (4) sexually active; (5) present sexual dysfunction identified by the female sexual function index with a score higher than 26.55 points; (6) present urinary incontinence (UI) identified by the international consultation on incontinence questionnaire (short form) with a score higher than 1 point; (7) never having performed exercises for the pelvic floor muscles; (8) never having participated of photobiomodulation treatment; (9) presenting a negative Pap smear result for cervical cancer precursor cells in the last year.

Exclusion criteria

We will exclude the women who are (1) unable to understand simple verbal commands or do not follow the instructions provided during the assessment; (2) affected by infectious or degenerative diseases; or (3) present III or IV genital prolapse according to the pelvic organ prolapse quantification.

Non-adherence criteria

We will consider as non-adherence criteria any report of unbearable pain during the gynecological evaluation, more than 20% absences during the intervention period, or those who withdraw their consent to participate in the research.

Interventions

The researchers will assign the women to one of the three groups: PFMT group, PFMT associated with active intracavitary photobiomodulation group, or PFMT associated with sham intracavitary photobiomodulation group. The exercise program will last 50 min per session, a 5-min warm-up, and a 45-min exercise, including the rest time between them. A total of 24 sessions will be conducted twice a week. Figure 1 illustrates the study flow diagram.

Figure 1 Flow diagram of the protocol.

There will be four exercise modalities in the PFMT group: (1) warm-up (fast and slow contractions, cough simulations, and plantar flexion associated with pelvic floor muscle contraction in a standing position; (2) pelvic floor muscle and respiratory training; (3) pelvic floor muscle and abdominal training (abdominal transverse muscle contraction associated with arm bridge and bipedal bridge); and (4) pelvic floor muscle training and pelvic mobility. There will be progressions for each modality related to the positioning, starting in dorsal decubitus and progressing to orthostatic or a harder level, such as using a ball to perform a bipedal bridge. There will be two series for the initial 4 weeks and three series for the last 4 weeks. During all the exercise modalities pelvic floor muscle contractions will be required. Figure 2 illustrates the protocol.

Figure 2 Protocol description illustration.

Created in Canva.

The PFMT group associated with photobiomodulation will perform the physical exercise, and right after exercising, the women will use the photobiomodulation device (Fluence Maxx-HTM®, Alpharetta, GA, USA) (Fig. 3). Table 1 demonstrates the LED equipment characteristics and irradiation parameters. The vaginal atrophy protocol will start with 25 J/cm2, continuous emission mode, during 450 s. A red, low-intensity light with a wavelength of 660 nm depicts the emission. The protocol is in accordance with previous studies (Lanzafame, de la Torre & Leibaschoff, 2019). The women will be in a lithotomy position during the photobiomodulation intervention. So, a probe covered by a nonlubricated condom but with a lubricant gel, will be inserted in the vaginal canal. Before starting the photobiomodulation intervention, the researcher will wear glasses and request the women to wear them.

Figure 3 Photobiomodulation device and intracavitary probe.

Table 1 Information about the LED equipment characteristics and irradiation parameters.

LED equipment	Irradiation parameters	
Equipment	Fluence Maxx-HTM®	
Wavelength (nm)	660 ± 10%	
Fluency (J/cm2)	0.0555	
Total power (W)	4	
Treatment time (s)	450	
Number of LEDs on probe	12	
Intimate applicator area (cm2)	72	
Irradiation	Intravaginal	

The women allocated to the PFMT associated with the sham intracavitary photobiomodulation group will perform the same protocol of the PFMT associated with the active intracavitary photobiomodulation group, except for the intracavitary photobiomodulation that will be off and will not release the red light. As the women will be wearing the protection glasses, they will not realize the absence of the red light.

For all groups, the interventions will be individualized. Furthermore, the women will receive educational content throughout the sessions informing them about menopause, risk factors, location and function of the pelvic floor, types of sexual dysfunctions, types of UI, functioning of the bladder and intestine, and the importance of a healthy lifestyle.

Outcomes

After the inclusion criteria, the included women will start the first evaluation before interventions. Then, there will be another evaluation after the twelfth and twenty-fourth session (midpoint and final evaluations). Another evaluation will occur 1 month after the end of the interventions (follow-up). The women will answer some questionnaires and perform physical and functional tests related to the pelvic floor muscles in accordance with the published license. Figure 1 illustrates the study flow diagram.

The primary outcome will involve measuring pelvic floor muscle vaginal manometry. The device Peritron, model 9300AV, will evaluate the pelvic floor muscle pressure. Initially, the evaluator will instruct the women to empty their bladder and remain in the lithotomy position. A conical sensor probe introduced in the vaginal canal will capture the pressure generated by the pelvic floor muscle. During the test, the evaluator will instruct the women to perform three maximum pelvic floor muscle contractions without contracting the abdominal, hip adductor, and gluteal muscles. The interval between the contractions will last 30 s. We will calculate the three attempts average and classify them according to the following criteria: very weak (7.5–14.5 cmH2O), weak (14.6–26.5 cmH2O), moderate (26.6–41.5 cmH2O), good (41.6–60.5 cmH2O) and strong (>60.6 cmH2O) (Angelo et al., 2017).

The international consultation on incontinence questionnaire – short form (ICIQ-SF) will evaluate UI; the female sexual function index (FSFI) will evaluate sexual function; the Utian Quality of Life (UQOL) scale will evaluate quality of life; the patient global impression of improvement (PGI-I) will evaluate the perception of improvement; the modified oxford scale will evaluate pelvic muscle strength; and the vaginal health index (VHI) scale will analyze vaginal atrophy. They will feature as secondary outcomes in the study’s evaluation.

The ICIQ-SF is a self-administered questionnaire that classifies urinary loss. The questionnaire is translated and validated in the Portuguese language (Tamanini et al., 2004). It presents four questions that evaluate the frequency, severity, and impact of UI on quality of life. It also presents a set of eight self-diagnostic items that allow evaluation of UI causes or situations experienced by patients. Only the first three questions will be scored and the total score will range from 0 to 21 points. The impact on quality of life will be classified as no impact (0 points); light impact (1 to 3 points); moderate (4 to 6 points); severe (7 to 9 points), and very severe (10 or more points) (Tamanini et al., 2004).

The FSFI is a self-administered questionnaire validated in the Portuguese language and Brazilian population. It presents nineteen questions about sexual activity during the last 4 weeks, involving six domains: desire, excitation, lubrication, orgasms, satisfaction, and discomfort/pain. The total score requires the domain’s sum. Scores below 26.55 points will classify as sexual dysfunction (Thiel Rdo et al., 2008).

The UQOL scale evaluates the quality of life, specifically for menopausal women. It is validated and adapted for Brazilian women and presents 23 questions divided into four domains: occupational, health, emotional, and sexual. The score spans from 23 to 115, with a higher score indicating a better quality of life (Utian et al., 2002; Lisboa et al., 2015).

The PGI-I is a simple and easy-to-use scale that presents clinical applicability to the perception of the intervention. It presents a single question about urinary functioning compared to the period before intervention. The score ranges from 1 (very much better) to 7 (very much worse) (Yalcin & Bump, 2003).

The modified Oxford scale evaluates the pelvic floor muscle through a manual palpation to measure the pelvic floor muscle strength. The scale presents six points to quantify the pelvic floor muscle strength according to the following criteria: grade 0: no contraction; grade 1: small and no sustained contraction; grade 2: low intensity and sustained contraction; grade 3: moderated contraction, feeling the intravaginal pressure compressing the evaluator’s finger; grade 4: satisfactory contraction, the evaluator’s fingers are compressed by the intravaginal pressure; grade 5: strong contraction, the evaluator’s fingers are compressed towards to pubic symphysis (Da Roza et al., 2013).

The VHI scale evaluates vaginal atrophy by assessing the vaginal humidity, vaginal liquid volume, vaginal elasticity, pH, and vaginal epithelium integrity. Values below 15 indicates that vaginal mucosa is atrophic. To evaluate the pH, we will position a universal pH tape (MERCK® (MColorpHastTM, Merck, Rahway, NJ, USA) in the right vaginal wall during 1 min without contact with the cervix or cervical mucus. A pH below and above five indicates normal and atrophy vaginal trophism, respectively (Bachmann & Nevadunsky, 2000). All the instruments that will measure the outcomes of the study are in accordance with the published license. Figure 4 illustrates the evaluation process.

Figure 4 Evaluation process.

Participants timeline

Initially, the included women will provide the information requested in the evaluation form, questionnaires, Oxford Modified Scale, vaginal manometry, vaginal health index, and vaginal pH. The midpoint evaluation will occur immediately after the twelfth session. The final evaluation will be conducted immediately after the twenty-fourth intervention. Another evaluation will take place 1 month after the last intervention (follow-up). The evaluations will follow the same protocol by a blinded evaluator. See the SPIRIT diagram (Fig. 5).

Figure 5 SPIRIT flow diagram.

Sample size

G-Power software version 3.1.9.2 calculated the sample size, according to the ANOVA F test repeated measures, within-between interaction, based on a previous study (Bezerra et al., 2020). We considered the data from the manometry results between groups after interventions. To calculate, we will consider a power of 80%, an alfa error of 5%, and an effect size of 0.23. Therefore, a total of forty-two individuals will participate in the study.

Recruitment

The women will be recruited at the urogynecology outpatient clinic located in the Maternity Hospital Januário Cicco School from the Federal University of Rio Grande do Norte, Natal, Brazil. The contact with the women will be through doctor referral, written disclosure, and personal contact.

Randomization, allocation concealment, and blinding

A numeric sequence generated by the software (https://www.random.org/) will randomize the women to allocate (1:1:1) three groups: PFMT group, PFMT associated with intracavitary photobiomodulation group, or PFMT associated with intracavitary photobiomodulation sham group. Each woman will present the same probability to allocate any group. An independent investigator who will not be involved in the evaluation and intervention process will perform it. The women and researchers involved in the assessments and interventions will be blinded.

Data collection

The data collection will occur at the urogynecology outpatient clinic located in the Maternity Hospital Januário Cicco School from the Federal University of Rio Grande do Norte, Natal, Brazil.

Statistical analysis

The software Jamovi (version 2.3.28) will analyze the data. The generalized estimated equations or mixed model will analyze the data at baseline, after the last intervention, and 1 month after the last intervention. We will choose the best distribution to represent the data, according to residuals Q-Q plot and histogram. Moreover, according to the Adjusted Consistency Index (A/C Index), Bias/Consistency Index (B/C Index), Quality/Consistency Index (Q/C Index), Chi-square and degrees of freedom (DF), and the intraclass correlation coefficient, we will evaluate the necessity of incorporating a fixed or random factor into the constant of the dataset. The link function will estimate the distribution of the dependent variable. Time, groups, and their interactions will represent the independent factors. The mean, mean difference, standard deviation, confidence interval, and p-value for all evaluation moments will represent the data. Bonferroni correction for multiple comparisons will calculate the post-hoc comparisons. Cohen’s d will calculate the effect sizes for all outcomes interactions. The significance level will be defined in all statistical testing as a p-value of less than 0.05. If there are any losses after randomization, we will perform an intention-to-treat analysis. If required, we will conduct imputation followed by sensitivity analysis, utilizing methods, such as last value carried forward and group means.

Data monitoring

This study will not involve participants in life-threatening or harmful interventions; therefore, it will not be necessary to create a data security monitoring board.

Harms

Adverse events will be carefully monitored during all study phases by asking the participants during and after each session if there is any adverse event related to pelvic floor muscle training and photobiomodulation. We will register and analyze any adverse event during the research period. Immediate interruption will occur in case of any risk. It is emphasized that the risks of this project are minimal, such as musculoskeletal pain or fatigue, itching, tingling, and burning sensations. The risks may vary depending on the group to which the volunteer is allocated. If any women withdraw, we will perform an intention-to-treat analysis or imputation data. In case of any harm related to the intervention, the researchers will provide all the necessary support.

Ethics and dissemination

The study will follow the Standard Protocol Items: Recommendations for Interventional Trials, Declaration of Helsinki, and resolution 466/12 of the National Health Council. The Ethics Committee of the Federal University of Rio Grande do Norte approved the study (n° 6.038.283). Also, The Brazilian Registry of Clinical Trials (ReBEC) registered the protocol (n° RBR-5r7zrs2). All the data will be stored at the physiotherapy department of the Federal University of Rio Grande do Norte with the principal investigator for 5 years, then they will be discarded.

The participants will be informed about the aims and procedures of the research. They will participate voluntarily, as determined by the 466/12 resolution of the National Council of Health. After that, if agree to participate, they will sign the informed consent form. Any personal information related to the participation in the study will not be shared during or after the study. After finishing the participation, the women randomized into the sham group will be invited to receive the photobiomodulation.

Discussion

This study protocol presents an innovative and feasible associated strategy to generate clinical improvements in women affected by urinary and sexual dysfunctions caused by GSM. In the literature, there is no protocol or intervention for using PFMT associated with photobiomodulation to treat women affected by urinary and sexual dysfunctions caused by GSM. So, it is possible that the association of PFMT with photobiomodulation might potentiate the improvements.

Although PFMT to treat women affected by GSM generates an increase in blood flow, speed of pelvic floor relaxation after a contraction, an increase of strength, a decrease in the tonus, and skin elasticity improvements (Mercier et al., 2020), the evidence reporting clinical improvements after PFMT for women affected by GSM are biased and scarce (Mercier et al., 2020, 2019). Photobiomodulation is an intervention used to treat several conditions, such as sexual dysfunction (Frederice et al., 2022), vulvovaginal disorders (Forret et al., 2023), and UI (De Marchi et al., 2023). There is evidence showing that photobiomodulation in the vaginal region increases collagen production, oxygenation, tissue tensile strength, vasodilator, accelerates wound healing, and reduces pain (Frederice et al., 2022; De Marchi et al., 2023). In this sense, photobiomodulation also might be a feasible strategy to treat women affected by urinary and sexual dysfunctions caused by GSM.

The association of photobiomodulation with other techniques, such as pilates (De Marchi et al., 2023) and stretching, (Frederice et al., 2022) has already been studied and presented improvements for pain, sexual dysfunction, and strength in women with pelvic and urinary dysfunctions (De Marchi et al., 2023; Forret et al., 2023). Considering these assumptions, due to women affected by GSM presenting alteration in the genitourinary tract, such as urinary and sexual dysfunctions (Palacios et al., 2017), it is highly possible that photobiomodulation potentiates the effects of PFMT in the genitourinary tract.

This protocol presents some limitations and strengths that must be acknowledged. There will be no sensor to confirm if the women are performing efficient pelvic floor muscle contractions during interventions. However, the researcher will stimulate the pelvic floor contraction during the whole session. Also, we will use validated, easy-to-use, and cheap instruments with clinical applicability to evaluate the effects of the treatments. Lastly, a long-term follow-up could demonstrate if the effects of the intervention would last longer.

Conclusion

PFMT associated with photobiomodulation protocol is expected to amplify the effects of isolated PFMT in women experiencing GSM accompanied by urinary and sexual dysfunctions. Also, we project that this combined protocol mirroring clinical practice is feasible for women. This preliminary study intends to explore the feasibility and initial trends for PFMT associated with photobiomodulation.

Supplemental Information

Supplemental Information 1 Test instruments information.

Supplemental Information 2 Global impression questionnaires for incontinence.

Supplemental Information 3 Oxford scale article.

Supplemental Information 4 SPIRIT Checklist: Recommended items to address in a clinical trial protocol and related documents.

Additional Information and Declarations

Competing Interests

Author Contributions

Human Ethics

Clinical Trial Ethics

Data Availability

Clinical Trial Registration

The authors declare that they have no competing interests.

Lívia Oliveira Bezerra conceived and designed the experiments, prepared figures and/or tables, authored or reviewed drafts of the article, and approved the final draft.

Maria Letícia Araújo Silva de Carvalho conceived and designed the experiments, prepared figures and/or tables, authored or reviewed drafts of the article, and approved the final draft.

Edson Silva-Filho conceived and designed the experiments, prepared figures and/or tables, authored or reviewed drafts of the article, and approved the final draft.

Maria Clara Eugênia de Oliveira conceived and designed the experiments, prepared figures and/or tables, authored or reviewed drafts of the article, and approved the final draft.

Palloma Rodrigues de Andrade conceived and designed the experiments, prepared figures and/or tables, authored or reviewed drafts of the article, and approved the final draft.

Maria Thereza Albuquerque Barbosa Cabral Micussi conceived and designed the experiments, prepared figures and/or tables, authored or reviewed drafts of the article, and approved the final draft.

The following information was supplied relating to ethical approvals (i.e., approving body and any reference numbers):

The Ethics Committee of the Federal University of Rio Grande do Norte approved the study (n° 6.038.283).

The following information was supplied relating to ethical approvals (i.e., approving body and any reference numbers):

The Brazilian Registry of Clinical Trials (ReBEC) registered the protocol (n° RBR-5r7zrs2).

The following information was supplied regarding data availability:

This is a registered report (a study protocol) so there is no data reported in this article.

The following information was supplied regarding Clinical Trial registration:

The Brazilian Registry of Clinical Trials (ReBEC) (RBR-5r7zrs2).

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
