# Peer review of "Pelvic floor muscle training associated with the photobiomodulation therapy for women affected by the genitourinary syndrome of menopause: a study protocol"

_PeerJ, doi:10.7717/peerj.17848_

## Round 0.1 · original submission · Major Revisions

Dear authors, at the moment your study does require significant revisions. Please, revise the reviewers reports and respond accordingly. Your re-submission may include data from a small cohort of test, to overcome the status of "study proposal".

Reviewer 1 ·

Basic reporting

This manuscript details a clinical study to evaluate the effects of PBMT in GSM patients either alone or in combination with pelvic floor exercises. The manuscript is generally well-written and well illustrated.

Experimental design

The general experimental design with 3 randomized groups is reasonable. The study will enroll 36 women randomized into 3 groups. The treatments and evaluations will occur over 8 weeks (16 treatment sessions). The sample size was apparently calculated to an 80% level. However, the overall sample size may be too low particularly if there is significant variance in individual patient response. Similarly, classical PBMT regimes use a QOD paradigm and this would equate to 3-4 sessions weekly. The regimen may not be optimal for PBMT. Treatment should be continued for a longer period of time, perhaps 3-4 months to allow sufficient time for tissue remodeling and for patients to reach a plateau in response.

Validity of the findings

The findings would add to understanding of the role of PBMT with 660nm in managing GSM. The proposed assessments are reasonable in order to evaluate treatment outcomes. However, the duration of treatment appears to be to short in this reviewer's opinion.

Additional comments

The study proposal has merit. However, revisions are warranted.

Reviewer 2 ·

Basic reporting

Dear authors, despite your interest in improving the quality of life of patients with genito-urinary menopausal syndrome, with new proposals, here are some comments below, right after the respective paragraph


1. Isolated treatments focused on the main causes of the dysfunctions, such as pelvic floor muscle training (PFMT) and photobiomodulation have shown significant improvements in genitourinary dysfunctions: PFMT improves the parameters of pelvic floor dysfunction, but is not supported by atrophy, the cause of postmenopausal changes. Photobiomodulation is still controversial

2. So, the association of PFMT with photobiomodulation may generate additional effects in the genitourinary area.:It would be more interesting to do it just with PBM to evaluate results from atrophy, as there is with microabaltive laser and radiofrequency as well. But the gold standard is still local hormone replacement

3. This study aims to create a PFMT protocol isolated and associated with photobiomodulation therapy in women affected by the genitourinary syndrome of menopause.In the same way, the evaluation of PFMT alone compared to the existing gold standards

4. We expect this protocol to demonstrate that the use of PFMT and photobiomodulation strategies is feasible and able to potentiate the recovery of women affected by the genitourinary syndrome of menopause: .PBM has no evidence for disorders arising from vaginal atrophy, evident in GSM, where the mucosa and submucosa suffer the effects of estrogen deficit.

5. Pelvic floor muscle training (PFMT) is a gold standard strategy to treat the GSM [4].: This is a paper of recommendations, PFMT is not the gold standard for menopausal atrophy

6 .Recently, photobiomodulation emerged as a new therapeutic approach to treat genitourinary dysfunctions :These 3 papers talk about dyspaureunia, mechanism of action and use in SUI

7. Considering these assumptions, photobiomodulation has been utilized to prevent muscle damage, repair muscle tissues, improve muscle performance, and reduce fatigue [12-15].: Muscular effects, none of them about atrophy of vaginal mucosa
8. The therapeutic effects generated by photobiomodulation have been associated with improvements in fatigue reduction, post-exercise recovery, and muscular performance [18,19,21].: None of them about PBM over mucosa
9. Considering the photobiomodulation physiological effects on the pelvic floor muscle tissue and the scarcity of clinical trials related to this subject, this study aims to create a PFMT protocol isolated and associated with the photobiomodulation therapy in women affected by the GSM presenting urinary and sexual dysfunctions.: Maybe a protocol with PFMT isolated?

10.. The protocol is in accordance with a previous study [23].:This study is not about mucosa It is about muscles

11.. It is emphasized that the risks of this project are minimal, such as musculoskeletal pain or fatigue, itching, tingling, and burning sensations.:The control group should not feel any kind of hot sensations, this could cause bias

12.So, it is possible that the association of a gold standard treatment, such as PFMT [4], : It is not a gold standard, it is a recommendation

13. there is little evidence reporting clinical improvements after PFMT for women affected by GSM : this conclusion is opposite to the previous one

Experimental design

no comment

Validity of the findings

1.In this sense, photobiomodulation also might be a feasible strategy to treat women affected by urinary and sexual dysfunctions caused by GSM.


Surely more data is needed for this claim.
This study should be based on more robust studies to suggest an interaction between PFMT and photobiomodulation
Maybe the effects of PBM over the vaginal mucosa rather than over muscles?

Additional comments

With the increase in women's life expectancy, more patients are seeking help, regarding changes in the pelvic floor, that cause dysfunction , affecting quality of life.
Well-studied techniques are necessary, since there are countless procedures in the medea that promise unproven improvement.
The evidence-based medicine is to prove what will really help these women.
This study should avoid comparing the use of photobiomodulation over the mucosa using data on muscles.

·

Basic reporting

No comment.

Experimental design

This paper is a study proposal, that is, a research project. The simple publication of this paper will not have scientific relevance, however the results of this project once carried out will help to elucidate some points that have not yet been clarified in the current literature. In this sense, it cannot be classified as original research but rather as a good study proposal.

The study proposal is good, however I would strongly suggest the inclusion of an intention-to-treat analysis and a review of the sample calculation carried out, in the current text the use of effect size is not justified.
In other words, the proposal still has points to be analyzed and improved.

Validity of the findings

Small impact validity, given that it is a study proposal. There are no conclusions, as the collections were not carried out. The collection schedule for this project is not presented.

---

## Round 0.2 · Major Revisions

Dear authors, thank your for your efforts.

This was a difficult decision. Please, refer to the reviewers comments.

While it is clear that everyone involved sees potential in the study's premise, the current design has been highlighted insufficient to produce reliable and meaningful results. Although, i do recognize, that it actually does reflect a lot of the 'real world' practice.

However, we need to care for the potential to result in the application of inadequate and unreliable treatments. In this case, because the current PBMT regimen of 2 times per week and the study duration may be insufficient to determine efficacy robustly.

If possible, try to revise again the study protocol (it can even be in a parallel way, comparative point of view) as per reviewer's suggestion. If revising the study design is not feasible due to logistical or financial constraints, consider rephrasing the study's objectives and scope. Emphasize that this is a preliminary study intended to explore the feasibility and initial trends, rather than a definitive efficacy study. Provide a stronger justification for the chosen regimen and duration. You can refer to any existing literature or pilot data that supports your current approach. Explain any practical or ethical constraints that prevent a longer or more frequent treatment regimen and how these constraints impact the study's scope. Emphasize the importance of your findings even if they are preliminary. Explain how your study contributes to the body of knowledge and can inform future, more comprehensive studies.

I hope this helps. Looking forward for your revised manuscript.

Reviewer 1 ·

Basic reporting

clear

Experimental design

The authors have increased the sample size. Unfortunately, the PBMT treatment regimen at 2X/wk is suboptimal and the duration of treatment is still too short in order to be able to determine efficacy. It is understood that the design requires in office treatment and that this incurs an expense. However, a poorly designed trial can result in erroneous findings and this does a disservice to patients and the literature. Treatment should be QOD (3 or 4X/wk) the duration of the study should be 3 months at a minimum, but ideally at least 4 months for the PBMT effects to reach a peak and plateau.

Validity of the findings

Unfortunately, the PBMT treatment regimen at 2X/wk is suboptimal and the duration of treatment is still too short in order to be able to determine efficacy. It is understood that the design requires in office treatment and that this incurs an expense. However, a poorly designed trial can result in erroneous findings and this does a disservice to patients and the literature. Treatment should be QOD (3 or 4X/wk) the duration of the study should be 3 months at a minimum, but ideally at least 4 months for the PBMT effects to reach a peak and plateau.

Reviewer 2 ·

Basic reporting

Dear Authors,
Thank you for clarifying the questions. All the comments are constructive.
1. "Therefore, it becomes crucial to investigate PBM alongside highly recommended therapies, as this could provide insights into an area that has not yet been extensively studied."
That is the first clinical trial, and precisely for this, the questions regarding its feasibility.

2.." However, in this case, we are considering the evaluation of pelvic floor muscles (PFM) as the primary outcome due to skeletal muscle weakness/atrophy, rather than skin atrophy".
Totally agree

4.; "we have revised the text to make it clearer and to avoid any doubt."
Thank you for that

5. "Thank you for your feedback; we have revised the text to make it clearer and to avoid any doubt. "
Thank you fot that

6. "We have also checked the order of the references."
Thank you for that

7. " We formulated the hypothesis for this study based on PBM on skeletal muscle. We are considering the evaluation of PFM as the primary outcome, focusing on skeletal muscle weakness/atrophy, rather than skin atrophy."
That has to be very clear, Thank you

9. "We have revised the manuscript to avoid any doubt"
Thank you for that

10. "We revised the manuscript and inserted a new reference."
Thank you for that

11. "We added the following clarification: The risks may vary depending on the group to which the volunteer is allocated"
great option

12. "We revised it in the text."
Very good!

13. "We revised the text and made it clearer"
Thank you for that

Experimental design

no comment

Validity of the findings

no comment

Additional comments

New technologies to improve QOL of our patients must follow scientific rigor , due to the risk of offering inadequate and unproven treatments.
I am satisfied with the authors' response

·

Basic reporting

No comment.

Experimental design

The justification for the impact factor used for the sample calculation was not presented.

Validity of the findings

No comment.

Additional comments

No comment.

---

## Round 0.3 · accepted · Accept

Dear authors, although controversial I am happy to let you know that I am accepting your work for publication in PeerJ. Thank you for your hard work.

Reviewer 2 ·

Basic reporting

Dear authors, I have no further comments.
You've made the necessary modifications

Experimental design

no comment

Validity of the findings

no comment

Additional comments

I have already made all the necessary comments and the authors adapted the study